# Mechanism for fluctuating pair density wave

Chandan Setty [1,2] ✉, Laura Fanfarillo [1,3] ✉ & P. J. Hirschfeld[1] ✉

In weakly coupled BCS superconductors, only electrons within a tiny energy window around the Fermi energy, $E_F$, form Cooper pairs. This may not be the case in strong coupling superconductors such as cuprates, FeSe, SrTiO$_3$ or cold atom condensates where the pairing scale, $E_B$, becomes comparable or even larger than $E_F$. In cuprates, for example, a plausible candidate for the pseudogap state at low doping is a fluctuating pair density wave, but no microscopic model has yet been found which supports such a state. In this work, we write an analytically solvable model to examine pairing phases in the strongly coupled regime and in the presence of anisotropic interactions. Already for moderate coupling we find an unusual finite temperature phase, below an instability temperature $T_i$, where local pair correlations have non-zero center-of-mass momentum but lack long-range order. At low temperature, this fluctuating pair density wave can condense either to a uniform d-wave superconductor or the widely postulated pair-density wave phase depending on the interaction strength. Our minimal model offers a unified framework to understand the emergence of both fluctuating and long range pair density waves in realistic systems.

Spatially uniform superconducting (SC) order formed from Cooper pairs with zero center-of-mass momentum is the energetically favored ground state in the conventional theory of Bardeen, Cooper and Schrieffer (BCS)[1]. Equivalently, the SC instability is signaled by a divergence in the static pair-fluctuation propagator, $L(\mathbf{q}, \Omega = 0)$, at $\mathbf{q} = 0$ once the pair instability temperature, $T_i$, is achieved[2]. On the other hand, a non-uniform order with non-zero center-of-mass momentum Cooper pair can occur when the divergence of the pair-fluctuation propagator is shifted to non-zero $\mathbf{q}$. First proposed by Fulde and Farrell (FF)[3] and independently by Larkin and Ovchinnikov (LO)[4], these solutions are stabilized in the presence of explicit time-reversal symmetry breaking from an external magnetic field. A modulated order parameter can also be realized in the presence of time-reversal symmetry where the spatial average of the gap vanishes. Termed pair-density waves (PDWs), these states are posited to exist in a variety of systems, including high-temperature cuprate superconductors (for a review, see ref. 5 and references therein).

While PDWs have been subject to much theoretical[6–19] and numerical[20–26] interest, a clear-cut analytically solvable model describing their origin from microscopic ingredients is lacking. From the experimental point of view, the interest for modulated pairing phases has been triggered by increasing experimental evidence for short-ranged PDW order in the underdoped region of the phase diagram of cuprates[26–38]. In particular,[32] reported the first clear observation via scanning tunneling spectroscopy of a vortex-induced PDW in Bi$_2$Sr$_2$CaCu$_2$O$_8$ at low temperature. More recent STM experiments provide further evidence in favor of a short-range PDW coexisting with the d-wave superconductivity in the SC phase and evolving into a PDW state in the pseudogap region[26,38]. This phase is characterized by a gap at finite temperatures but lacks long-range order, and can be characterized as a "fluctuating pair density wave", locally pinned by disorder. Such a state also provides an explanation for many other experimental signatures of the cuprates, including the existence of vestigial charge density wave order arising from partial melting of a PDW[5,15,39]. However, there is currently no microscopic model supporting this picture. Hence it is urgent to seek a unified framework that subsumes both fluctuating and long-range ordered PDW phases under a single paradigm by providing a concrete description of their origin.

In this work, we show that a Fermi liquid subjected to a finite anisotropic interaction is unstable toward a modulated SC phase in the

[1]Department of Physics, University of Florida, Gainesville, FL, USA. [2]Department of Physics and Astronomy, Rice Center for Quantum Materials, Rice University, Houston, TX 77005, USA. [3]Scuola Internazionale Superiore di Studi Avanzati (SISSA), Via Bonomea 265, 34136 Trieste, Italy. ✉e-mail: csetty@rice.edu; laura.fanfarillo@gmail.com; pjh@phys.ufl.edu

strong coupling limit. Whether this phase is a 'fluctuating' PDW (FPDW) or long-range order PDW is determined by temperature as well as the coupling strength defined by the ratio $\alpha = E_B/E_F$, with $E_F$ the Fermi energy and $E_B$ the bound state energy for pair formation.

Our strategy is to solve the self-consistent gap equation for the homogeneous $d$-wave superconductor and analyze the momentum dependence of the SC fluctuations. The expansion of the static pair propagator $L_\mathbf{q}$ in powers of momentum transfer $\mathbf{q}$, can reveal, in fact, critical fluctuations of Cooper pairs with finite center-of-mass momentum, that makes the homogeneous solution unstable toward a modulated SC phase. Since we study the action only to second order in the pair field we cannot distinguish between an instability to a FF or LO state. We observe the emergence of a modulated SC state already at intermediate coupling $\alpha \sim 0.7$. The appearance of such a state is linked to the existence of fluctuating terms that lower the momentum rigidity of the Cooper pairs. These terms directly follow from the anisotropy of the pairing interaction that affects the momentum dependence of the pairing susceptibility *already* in the normal phase.

Our results are summarized in the phase diagram, Fig. 1. $T_i$ is the instability temperature of the homogeneous $d$-wave state obtained within the mean-field approximation. The analysis of fluctuations allows us to define two different regimes. At weak coupling, $\alpha \ll 1$, the uniform $d$-wave paired state is the ground state; at larger $\alpha$ (strong coupling), SC fluctuations at finite momentum lead to two modulated pairing phases— the $T = 0$ PDW ground state and a higher temperature FPDW phase that condenses into a PDW ordered phase below a coherence temperature ($T_c$). The strong and weak coupling regions are separated by the line $T = T^*$. This is the temperature at which the finite momentum fluctuations around the homogeneous solution become critical. As expected in the BCS limit, the instability temperature $T_i$ and the coherence temperature $T_c$ coincide at weak coupling. In the strong coupling regime, we anticipate that $T_i$ and $T_c$ decouple since Cooper pairs are formed but with no long range coherence. In this work, we do not perform here any calculation of the coherence temperature $T_c$ inside the modulated phase. However, in analogy with results obtained for homogeneous $s$-wave superconductors in the strong coupling limit[40], we expect that $T_c < T_i$ for $\alpha > 1$ as well. The FPDW is found for temperature $T_c < T < T_i$ and it is characterized by pairs with finite

momentum with no coherence. At $T = 0$, the ground state can be either the uniform $d$-wave solution or the long-range PDW depending on the value of $\alpha$. Hence our model captures two key experimentally postulated modulated Cooper phases —a FPDW and a long-range PDW—in a single unified scheme.

The mechanism we present in this paper predicts spatially modulated pairing phases for $\alpha = E_B/E_F > 1$, i.e., in strongly coupled electronic systems, with anisotropic interactions. Examples of low-density electronic materials include the Fe-based superconductor FeSe where quantum oscillations[41] as well as transport and scanning tunneling spectroscopy[42] show that both the electron and hole pockets are tiny with Fermi energies comparable or even smaller than the SC gap and for which we find several proposals of BCS-BEC cross-over physics in the literature[43–45]. Other "mixed-band" superconductors such as O vacancy- or Nb-doped SrTiO$_3$ have one partially filled band with a large Fermi surface while the Fermi level intersects the other at or close to the band bottom[46]. Even if these materials typically have more than one band close to or crossing the Fermi level, the results from our minimal model may eventually provide a suitable starting-point for the analysis of possible instabilities toward modulated pairing states in dilute multiorbital superconductors. Our results may also be relevant to the recent observation of superconductivity in twisted-bilayer graphene[47] where interactions can be large compared to the bandwidth leading to large inter-particle distances[48] and hence possible strongly-coupled Cooper pairing.

The modulated phases we propose in this work, that include both the long-range ordered PDW as well as the FPDW at finite temperature, are distinct from earlier proposals in literature. Loder et al.[10], considered similar models characterized by nearest neighbor attractive interaction with $d$-wave symmetry and found Cooper pairing with finite center-of-mass momentum above a critical interaction strength. In refs. [18, 19], a modulated superconducting state is found in models which have correlated pair-hopping interactions. Other models that admit modulated SC ground states were proposed in the context of cold atoms[9] where local interactions were considered in systems with multiple bands. Those references focused on the analysis of the long-range ordered state (mainly at zero temperature) without exploring the FPDW phase. The key contribution of our work is it provides an analytically tractable model where both fluctuating and long-range ordered PDWs can be explained under a single unified framework.

## Results
### Model
Let's consider a single band SC system. The kinetic part of the Hamiltonian reads $H_0 = \sum_{\mathbf{k}\sigma} \xi_\mathbf{k} c_{\mathbf{k}\sigma}^\dagger c_{\mathbf{k}\sigma}$, where $\xi_\mathbf{k} = \epsilon_\mathbf{k} - \mu$, $\mu$ is the chemical potential, $\epsilon_\mathbf{k} = \mathbf{k}^2/2m$ the parabolic dispersion and we further assume $2m = 1$. The pairing interaction is given by

$$H_I = -g \sum_\mathbf{q} \theta_\mathbf{q}^\dagger \theta_\mathbf{q}, \tag{1}$$

$g$ is the constant SC coupling and $\theta_\mathbf{q}$ is defined as

$$\theta_\mathbf{q} = \sum_\mathbf{k} f_{\mathbf{k},\mathbf{q}} \, c_{-\mathbf{k}+\frac{\mathbf{q}}{2},\downarrow} c_{\mathbf{k}+\frac{\mathbf{q}}{2},\uparrow}. \tag{2}$$

where $f_{\mathbf{k},\mathbf{q}} = (h_{\mathbf{k}-\mathbf{q}/2} + h_{\mathbf{k}+\mathbf{q}/2})/2$ is a form factor. In this work $h_\mathbf{k}$ can be any anisotropic form factor; we consider, e.g., $h_\mathbf{k} = (k_x^2 - k_y^2)/\Lambda$ with $d$-wave form. The pairing energy scale is $\Lambda$ i.e., the high energy cut-off set by the inverse lattice spacing and much larger than $E_F$. Our results do not depend qualitatively on the exact form of the anisotropy, provided it is strong enough, but they are distinct from the conventional $s$-wave case $f_{\mathbf{k},\mathbf{q}} = 1$. Note that the interaction Hamiltonian we choose above already assumes an attractive pairing interaction and does not begin from any repulsive Hubbard-type model. Nevertheless, the mechanism

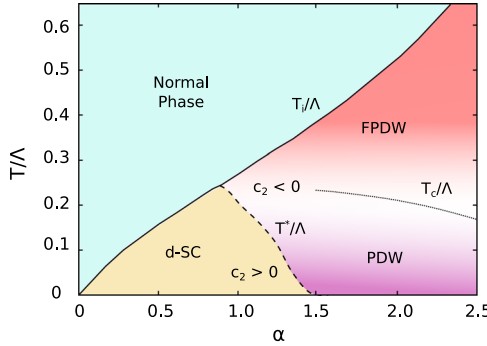

**Fig. 1 | Phase diagram for $\alpha = E_B/E_F$ vs $T$.** The instability temperature for the $d$-wave superconductor, $T_i$, defines the transition from a Fermi liquid (light blue) to the SC state. At weak coupling the pairing state is a homogeneous $d$-wave superconductor (gold). Increasing $\alpha$ the system develops critical fluctuations at finite momentum and the $d$-wave SC state becomes unstable toward a non-homogeneous SC state (pink and purple regions). $T^*$ is the temperature at which the momentum rigidity parameter $c_2$ vanishes. The fluctuating PDW (pink) condenses below a coherence temperature $T_c$ into a long-range ordered state that can be an homogeneous $d$-wave SC state (gold) or a PDW (purple) depending on the coupling strength, schematically represented by a solid line. $T_c$ coincides with $T_i$ at weak coupling while at strong coupling it is expected that $T_c < T_i$[40]. Note that the actual instability temperature of the FPDW, $\bar{T}_i$, is somewhat higher than $T_i$ (see Supplementary Note 4). Temperatures are renormalized by the energy range of the pairing, $\Lambda$ that is the largest energy scale of our model.

we present is 'microscopic' in the sense that the order parameter can be evaluated over the entire phase diagram in terms of the microscopic parameters such as Cooper binding energy and Fermi energy. Such a treatment is distinct from Ginzburg-Landau theory and consistent with terminology used in previous literature[49,50].

We use the standard Hubbard-Stratonovich transformation to decouple the interaction term, Eq. (1) and to derive the effective action in term of the bosonic pairing field $\Delta$ (for a detailed derivation see Supplementary Note 1).

## Pair susceptibility

In standard BCS superconductors, the mean-field value of the pairing field is defined by minimizing the action with respect to the homogeneous $\mathbf{q} = 0$ value of $\Delta$ and then solving this equation together with the one for the chemical potential. To study fluctuations of the pairing field around the mean-field value, we analyze the gaussian action obtained by retaining up to the second order in the fluctuating field with arbitrary momentum $\mathbf{q}$ given by

$$S_G[\Delta_{\mathbf{q}}] = \sum_{\mathbf{q}} L_{\mathbf{q}}^{-1} |\Delta|_{\mathbf{q}}^2. \qquad (3)$$

where $L_{\mathbf{q}}^{-1}$ is the static pairing susceptibility. As discussed in[51] and in the Supplementary Notes 1–3 of this work, we need to consider in principle both the real and imaginary part of the fluctuating field. However, here we are interested in the analysis of the static limit of the fluctuation, for which real and imaginary part are decoupled. Hereafter we will analyze the gaussian fluctuation of the amplitude mode only to investigate the possible emergence of a spatially modulated pairing fluctuation out of a homogeneous d-wave SC state. The explicit form of the static pairing susceptibility for the amplitude mode is $L_{\mathbf{q}}^{-1} = g^{-1} + \Pi_{\mathbf{q}}$, where the particle-particle propagator reads

$$\Pi_{\mathbf{q}} = \frac{T}{V} \sum_{\mathbf{k}n} \frac{(i\omega_n + \xi_{\mathbf{k}+\mathbf{q}})(i\omega_n - \xi_{\mathbf{k}}) - f_{\mathbf{k},0} f_{\mathbf{k}+\mathbf{q},0} \Delta^2}{(\omega_n^2 + E_{\mathbf{k}}^2)(\omega_n^2 + E_{\mathbf{k}+\mathbf{q}}^2)} f_{\mathbf{k},\mathbf{q}}^2. \qquad (4)$$

with $E_{\mathbf{k}}^2 = \xi_{\mathbf{k}}^2 + f_{\mathbf{k},0}^2 \Delta^2$. Here $T$ is the temperature and $V$ the volume. Note that since $2m = 1$, energies have dimensions of 2-D $V^{-1}$, and $L_{\mathbf{q}}^{-1}$ is therefore dimensionless.

The static susceptibility can be expanded in the hydrodynamic limit as

$$L_{\mathbf{q}}^{-1} = c_0 + c_2 q^2. \qquad (5)$$

The instability temperature is defined as the highest temperature at which the susceptibility diverges, i.e., $c_0 = g^{-1} + \Pi_0|_{T = T_i} = 0$, as we assume that the minimum of the action, Eq. (3), is associated with the homogeneous order parameter. The coefficient $c_2 = (\partial^2 L_{\mathbf{q}}^{-1} / \partial q^2|_{q=0})/2$ provides instead information about the momentum rigidity of the fluctuating Cooper pairs i.e., the energy needed to move the center-of-mass momentum of the Cooper pairs from zero to a finite value. A negative momentum rigidity, $c_2 < 0$, implies that finite momentum fluctuations can lower the energy of the system making the homogeneous SC solution unstable. This means that the highest temperature at which the pairing susceptibility, Eq. (5), diverges is actually associated to a critical mode with finite momentum.

In what follows we analyze the momentum-dependence of the static susceptibility, Eq. (5), looking for a sign change of the momentum rigidity parameter $c_2$ and using it as a proxy to identify possible spatially modulated SC regions in the phase diagram. It is worth noticing that $c_2$ is directly affected by the momentum properties of the pairing susceptibility i.e., the pairing symmetry. From Eq. (4), it is easy to verify that the anisotropy of the interactions affects the momentum dependence of the propagator not only in the SC phase via the symmetry of the SC order parameter, but also above the instability

temperature $T_i$ where $\Delta = 0$ due to the overall form factor $f_{\mathbf{k},\mathbf{q}}^2$ at the numerator. This reflects in a strong momentum dependence of the contributions to the rigidity parameter depending on the symmetry of the pairing interaction. We discuss below how this affects the development of critical finite-momentum fluctuations.

The mean-field analysis for the homogeneous d-wave superconductor is shown in Fig. 2. In panels (a)–(b) we report the self-consistent numerical mean-field solutions for the pairing function $\Delta$ and the chemical potential $\mu$ as a function of temperature $T$ for three representative cases of the pairing strength $\alpha = E_B/E_F = 0.5, 1.0, 2.0$, where for simplicity the weak-coupling expression $E_B = \Lambda e^{-2/g}$ is used at all $\alpha$. In panels (c)–(d) we show the same mean-field results at $T = T_i$ and $T = 0$ as a function of $\alpha$. The change of sign of the chemical potential with increasing coupling strength is well-known from the BCS-BEC crossover problem[52–56]. In the weak-coupling regime, the pairs are loosely bound and we recover the BCS expression $\mu \sim E_F$. As the interaction increases, all fermions strongly bind in pairs and $\mu$ becomes negative and proportional to $-E_B$. In both the weak and strong coupling limits, the curves are similar to those derived for s-wave superconductors in[56], showing that the d-wave symmetry of the pairing interaction does not affect the mean-field results qualitatively.

We first study the SC fluctuations above the instability temperature by analyze the static pairing susceptibility in the hydrodynamic limit, Eq. (5). The mass term $c_0$ is positive and vanishes as the temperature approaches the instability temperature as expected from a Ginzburg-Landau description of the transition.

The analysis of the momentum rigidity of the fluctuating pairs above $T_i$ is shown in Fig. 3. The weak coupling region is characterized by a standard regime of fluctuations with $c_2 > 0$. Here Cooper pairs with zero center-of-mass momentum are stable. Increasing $\alpha$, the momentum rigidity for the d-wave pairing interaction (continuous line) monotonically decreases and becomes negative at intermediate coupling, $\alpha > 0.7$, as shown in Fig. 3(a). This means that finite momentum critical fluctuations grow, increasing the coupling strength up to a critical value of the interaction for which the homogeneous SC solution can become unstable toward a modulated phase. Notice that $c_2$ becomes very small and eventually changes sign in the crossover between weak and strong coupling where also the chemical potential changes sign from positive to negative, see inset Fig. 3(a). The result changes qualitatively for the isotropic s-wave interaction (dashed line) where the rigidity parameter decreases but remains positive even at strong coupling for the set of model parameter of our study (This result differs from the analysis of[55], in which a sign change of the rigidity parameter is found for a SC system with s-wave pairing symmetry at strong coupling. In this case, however, the authors first perform a strong coupling expansion of the pairing susceptibility and only subsequently expand the approximated result in powers of $q$.).

To characterize the modulated SC state and check its stability, we expand the static susceptibility to higher order in momentum

$$L_{\mathbf{q}}^{-1} = \sum_{n} c_n \mathbf{q}^n, \quad \text{with} \quad c_n = \frac{1}{n} \frac{\partial^n L_{\mathbf{q}}^{-1}}{\partial \mathbf{q}^n}\Big|_{\mathbf{q}=0} \qquad (6)$$

We report the coefficients of the momentum expansion at $T_i$ in Fig. 3(b). Results are shown as a function of $\alpha$ for the coupling regime in which $c_2 \lesssim 0$. We need to expand the susceptibility up $n = 6$ to find $c_6 > 0$, since for our set of model parameters $c_4 < 0$ as in the conventional BCS case.

We analyze the momentum dependence of the static susceptibility at $T_i$ in Fig. 4, where we show the expansion of Eq. (6) up to sixth order for different values of $\alpha$. At the instability temperature, $c_0 = 0$ by definition and the minimum of the function is determined by the higher order coefficients. At weak coupling, where $c_2$ is large and positive, the minimum of $L_{\mathbf{q}}^{-1}$ is located at zero momentum. As the pairing interaction increases $c_2$ becomes small and eventually changes

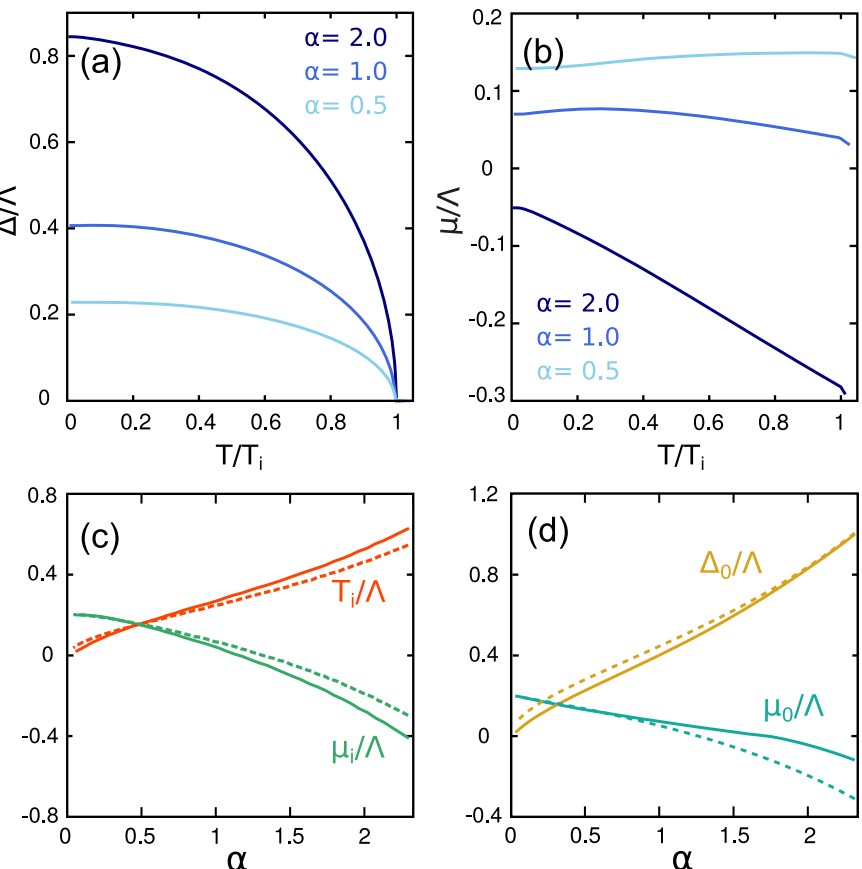

**Fig. 2 | Mean-field results for the spatially homogeneous *d*-wave super-conductor. a,b** Self-consistent solutions of the pairing order parameter $\Delta(T)$ and the chemical potential $\mu(T)$ for three representative values of $\alpha$. Temperatures are normalized to the instability temperature $T_i$ defined as the temperature at which the static pairing susceptibility $L_{q=0}$ diverges, while $\Delta$ and $\mu$ are scaled with $\Lambda$. **c** Instability temperature $T_i$ and chemical potential $\mu_i \equiv \mu(T = T_i)$ as a function of $\alpha$. **d** $T = 0$ solutions: $\Delta_0 \equiv \Delta(T = 0)$ and chemical potential $\mu_0 \equiv \mu(T = 0)$ as a function of $\alpha$. For comparison we show also the results of the isotropic s-wave case in dashed lines. Computations are performed using $\Lambda = 11$, $E_F = 2.2$ in units of $2m = 1$.

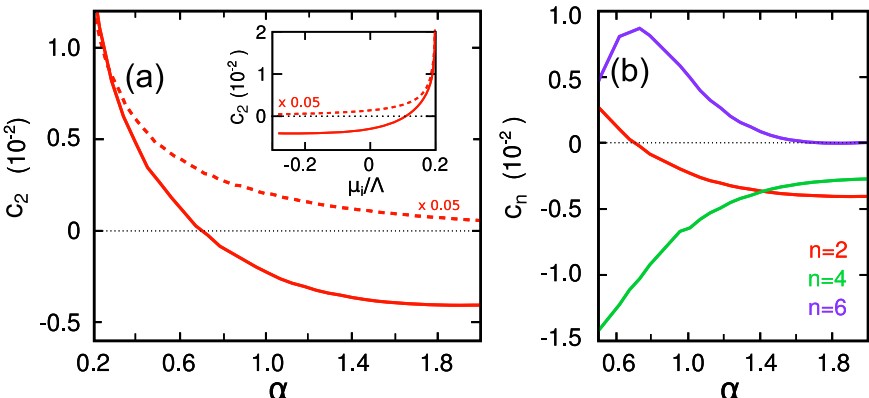

**Fig. 3 | Coefficients of the momentum-expansion of the static susceptibility as a function of the coupling strength $\alpha = E_B/E_F$ at $T = T_i$. a** The momentum rigidity $c_2(\alpha)$ for *d*-wave (solid line) and *s*-wave pairing interaction (dashed line). In the anisotropic *d*-wave case $c_2$ becomes negative at intermediate coupling, $\alpha \sim 0.7$ indicating that the homogeneous *d*-wave SC is unstable. Inset: $c_2(\mu_i)$, the sign change of the momentum rigidity occurs around the same range in which $\mu_i$ turns from positive to negative values. The momentum rigidity for the isotropic s-wave case remains positive regardless the coupling strength. **b** $c_n(\alpha)$ coefficients, $n = 2, 4, 6$, for the *d*-wave pairing. The positive value of $c_6$ allows to recover the stability of the action. The computation of the higher order coefficients allows to define the finite momentum of the critical mode and the relative instability temperature. We use here the same set of parameters of Fig. 2 and plot the results in dimensionless units i.e., $c_n \equiv c_n \Lambda^{n/2}$.

sign at $\alpha \sim 0.7$. Here, since $c_4 < 0$, the minimum shifts discontinuously to a finite momentum $\bar{Q}$, i.e., by increasing the interactions the modulated phase emerges at $T_i$ via a first order transition from the homogeneous *d*-wave SC solution, in analogy with the results found at

$T = 0$ in[10,18]. The non-zero value of $\bar{Q}$ at $\alpha \sim 0.7$ signals the formation of the FPDW state with finite momentum pairing but no long range coherent order. Note that the finite order parameter jump $\bar{Q}$ is a non-universal quantity and depends on microscopic details of the chosen

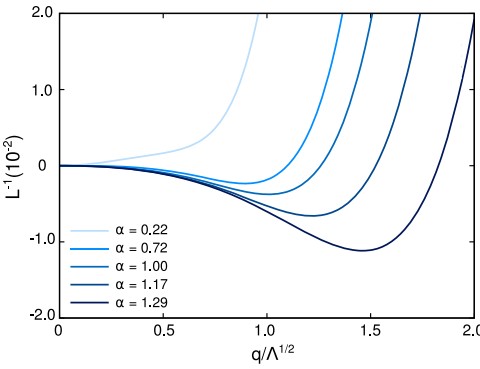

**Fig. 4 | Momentum dependence of the sixth order expansion of $L^{-1}$ at $T_i$ (dimensionless units).** At weak coupling, $\alpha = 0.22$, we find the homogeneous $d$-wave SC. The momentum rigidity $c_2$ is large and positive, and the minimum of the inverse of the susceptibility is at $q = 0$. At intermediate coupling, $\alpha \sim 0.7$, $c_2$ vanishes and the minimum of $L^{-1}$ appears at a finite $\bar{Q}$ of order 1. Same set of parameters of Fig. 2.

model, as is a feature of any generic first order transition. The momentum characterizing the modulated phase shifts toward larger values increasing the coupling parameters. In the strong coupling regime, $\alpha \gg 1$, the minimum occurs at $q/\sqrt{\Lambda} \gg 1$, (not shown), for this range of the interaction the analysis of the momentum characterizing the modulated phase requires the implementation of a non perturbative approach.

To further prove the stability and quantitatively characterize the modulated phases we expand the polynomial form, Eq. (6), around its minima $\bar{Q}$ so that we can define the susceptibility of the modulated SC phase

$$L_{\mathbf{q}}^{-1} \sim \bar{c}_0 + \bar{c}_2(\mathbf{q} - \bar{Q})^2. \tag{7}$$

Here $\bar{c}_0$ and $\bar{c}_2$ are the mass and the momentum rigidity parameter associated with the modulated SC phase. In the Supplementary Notes 2, 3, 4, we show that $\bar{c}_0 = a(T - \bar{T}_i)$ with $\bar{T}_i = T_i + \delta T$ and, $\delta T > 0$, and that $\bar{c}_2 > 0$, thus demonstrating stability of the modulated phases.

The sign change of the momentum rigidity parameter discussed at $T = T_i$ can be traced down in temperature (dashed line in Fig. 1). At $T = 0$ the homogeneous $d$-wave state becomes unstable, now toward a PDW, for a slightly higher value of the coupling where the chemical potential $\mu$ also changes sign (see Fig. 2d). The stability of the PDW phase requires expanding up to the sixth-order, $c_4 < 0, c_6 > 0$ as we show in the Supplementary Notes 2–4.

The results of our numerical study are summarized in the phase diagram of Fig. 1. We characterized the SC region below $T_i$ by the sign of the momentum rigidity parameter (dashed line). The sign change of the $c_2$ coefficient at strong coupling signals the presence of critical SC fluctuations at finite momentum that make the $d$-wave homogeneous state unstable toward either an FPDW or PDW. The pink and purple regions indicate the FPDW and the long-range ordered PDW state at high and low temperatures respectively. We leave for future work the explicit calculation of the coherence temperature below which the FPDW condenses. The color gradient indicates approximately the expected $T_c(\alpha)$ behavior based on previous analysis of the coherence energy scale for the homogeneous $s$-wave SC state[40].

Analytical calculations of the momentum rigidity can be easily performed within a simplified model in which the chemical potential is used as parameter. Both at $T_i$ and $T = 0$, we find qualitatively the same results discussed within the numerical study. In particular, within the analytical calculations sketched in the Supplementary Note 5, the momentum rigidity parameter follows the chemical potential

behavior, i.e., $c_2(\mu) < 0$ for $\mu < 0$. This relation is qualitatively in agreement with the numerical study performed computing self-consistently $\mu(\alpha)$, as one can see from the inset of Fig. 3a.

The strategy implemented here to investigate how finite momentum fluctuations become critical at strong coupling is based on the analysis of the momentum rigidity parameter. This method presents two main advantages with respect to other theoretical approaches. On the one hand, as already discussed, it allows us to explore the finite temperature regime and analyze the FPDW state. On the other, it provides a physical understanding of the importance of the anisotropy of the pairing interactions in the development of the modulated phase. As one can see in Eq. (4), the symmetry of the pairing interactions dramatically affects the momentum dependence of the propagator not only in the SC phase, but also in the normal one when $\Delta = 0$ due to the overall form factor $f_{\mathbf{k},\mathbf{q}}^2$. This is reflected in a strong momentum dependence of the contribution to the momentum rigidity parameter. In fact, after performing analytically the Matsubara summation, the computation of the $c_2$ coefficient reduces to an integral over the Brillouin zone $c_2 = \frac{1}{V}\sum_{\mathbf{k}} I_2(\mathbf{k})$. The expression for $I_2$ is given in the Supplementary Note 3, but here we show here in Fig. 5 2D maps of $I_2(\mathbf{k})$ for both $s$-wave and $d$-wave at $T = 0$ and $T = T_i$. In the isotropic $s$-wave case, the contributions to the momentum rigidity coming from different momenta, $I_2(\mathbf{k})$, are positive at any $(k_x, k_y)$. Whereas, in the $d$-wave case the contributions to the momentum rigidity coming from the nodal regions are negative and dominate the overall sign of the $c_2$ coefficient.

## Discussion

A consistent explanation for the occurrence of both static and fluctuating Cooper pairs with finite momentum in the phase diagram of materials such as cuprates has been a long-standing problem. This is primarily because an identification of the microscopic ingredients driving such exotic pairing has been elusive. The results in this paper point toward a simple and unified framework that naturally promotes both fluctuating and static pair-density wave (FPDW and PDW) phases over their zero momentum counterparts. Figure 1 summarizes the main conclusions of our work, supported not only by numerical evaluations but also transparent analytical estimates (see Supplementary Note 5). The two key ingredients resulting in a high temperature FPDW and low temperature PDW phases are (a) anisotropic (e.g., $d$-wave) pair interactions and (b) intermediate to strong coupling ratio of $\alpha = \frac{E_B}{E_F}$, where $E_B$ is the pair binding energy for two electrons on the Fermi surface in the presence of an attractive interaction, and $E_F$ is the Fermi energy. For the specific set of parameters presented here, below a critical value of $\alpha \sim 0.7$, only uniform zero momentum $d$-wave pairing is favored. In the approximate range of $0.7 \lesssim \alpha \lesssim 1.5$, the FPDW phase, characterized by a negative momentum rigidity $c_2$ and positive $c_6$ (see Fig. 3), is stable over a range of temperatures below the instability temperature $T_i$. However, in this range of $\alpha$ a uniform $d$-wave pair is still favored at zero temperature. For $\alpha \gtrsim 1.5$, the PDW phase is more stable than a uniform solution at $T = 0$ and a finite momentum pair exists for all temperatures below $T_i$. The modulation wave vector $\mathbf{Q}$ of the paired phases is determined by the ratio $\alpha$ and acquires a jump with increasing $\alpha$ as in a first order transition (see Fig. 4).

It is important to note that while in our paper the critical value of $\alpha$ for which $c_2$ changes sign appears to be of order unity, which is nominally outside the range of standard BCS weak-coupling theory, we emphasize that this value is a finite non-zero number that does not take a universal value. Depending on parameters, we can easily produce critical values of $\alpha$ of order 0.3, which might be considered weak-coupling (see Supplementary Fig. 2). Hence, there appears to be no physical reason that necessarily constrains FPDW and PDW phases to be in the strong coupling regime of $\alpha \sim 1$.

Furthermore, we note the key role of the existence of a lattice momentum cut-off. If the cut-off were taken to infinity, the modulated

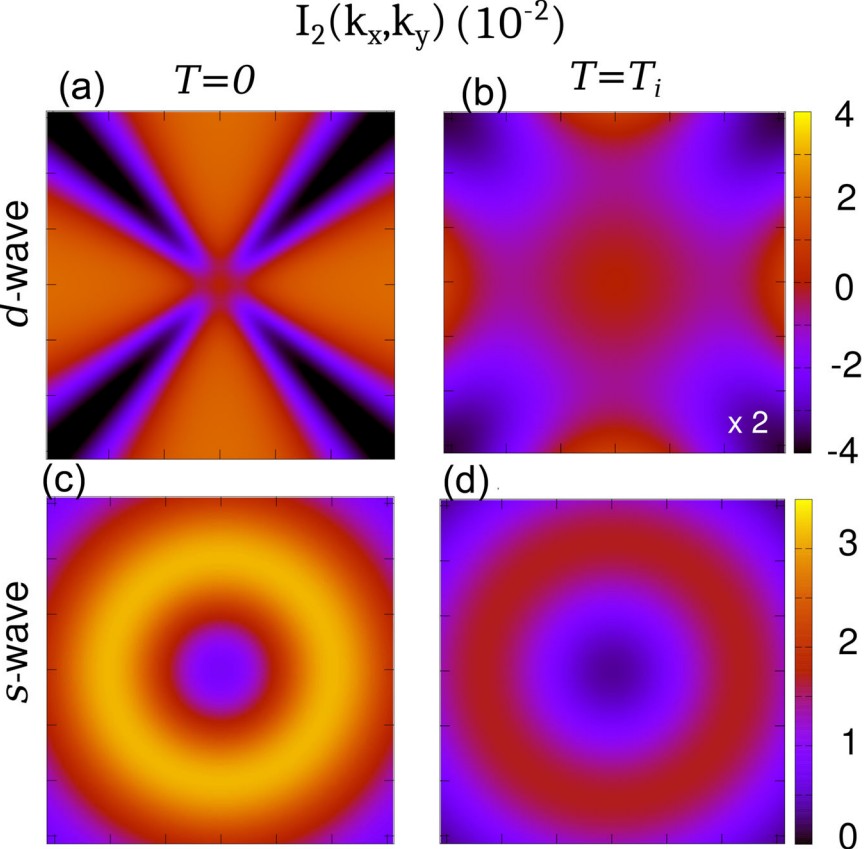

**Fig. 5 | Behaviour of the integrand $I_2(k)$ in the Brillouin zone.** $I_2(k_x, k_y)$ color maps at $T = 0$ and $T = T_i$ and $\alpha = 2.0$ for the $d$-wave (($\mathbf{a}$, $\mathbf{b}$)) and $s$-wave (($\mathbf{c}$, $\mathbf{d}$)) case. The anisotropy of the interactions affects the momentum dependence of the propagator both in the SC and normal phase. This reflects in a strong momentum dependence of the contribution to the momentum rigidity parameter. For the $d$-wave case negative contributions to the rigidity are found both at $T = 0$ and $T = T_i$ from $k$-points close to the nodal region. We use the same set of parameters of Fig. 2 and plot the result in dimensionless units for momenta $|k_i|/\sqrt{\Lambda} < \pi$, with $i = x, y$.

solution vanishes as is outlined in the Supplementary Note 5. This is further highlighted in the recent exact two-body and variational wave function solution[57] where it was shown that in the absence of a cut-off in the interaction, the modulated solution on a lattice loses to the homogeneous solution.

Recently refs. 58, 59 discussed the possible existence of a diagonal pair density wave order in the cuprates. To obtain a diagonal pair density wave order in our analysis, the interaction must contain a dominant $B_{2g}$ pair-fluctuation interaction. This would imply that the form factor of the kind $\sin k_x \sin k_y$ would be dominant in the fluctuations instead of the $\cos k_x - \cos k_y$ form factor in Eq. (1). With this form factor, our result continues to hold but with the dominant instability now occurring along the diagonals. Note, however, that our theory does not justify which of the two ($B_{1g}$ or $B_{2g}$) symmetry channels are the dominant fluctuations. In either of the two cases, one can obtain the finite momentum pairing instability without requiring a strong coupling treatment of the theory.

The FPDW and PDW phases are stabilized by contributions to the fluctuation free energy arising from momenta close to the nodal regions in the Brillouin zone. These contributions, which also should drive strong anisotropy in the phase stiffness near $T_i$, are suppressed (enhanced) at weak (strong) coupling thus leading to a modulated phase above a critical pairing strength. This simplified picture is confirmed from our numerical calculations (Fig. 5). Finally, while our work primarily focuses on the instability temperature $T_i$ in the strong coupling limit, the behavior of the condensation temperature $T_c$ and the fluctuations around the PDW ground state in this setting are open problems that will require further investigations. Our work does not consider the competing effects of a nematic superconducting phase that has been

phenomenologically found to suppress the PDW at $T > 0$ in 2D[7,11]. In addition, even if allowed by our model, we have not addressed the possible coexistence at low $T$ of a PDW and a homogeneous $d$-wave superconductor, as suggested by cuprate experiments[5,26]. Our results as such set the stage for future microscopic descriptions of modulated superconductivity in strongly coupled materials.

Note added: During review of the manuscript, another work based on repulsive interactions yielding PDWs became available[60].

## Methods

We used standard many-body field theoretic methods for all computations. Details of these methods are included in the Supplementary Information provided.

## Data availability

All data generated or analyzed during this study are included in this published paper (and its Supplementary Information files)

## Code availability

All codes used to generate or analyze the results of this study are available from the corresponding author (C.S.) on reasonable request.

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

## Acknowledgements
We thank P. Abbamonte, B.M. Andersen, S. Caprara, A. Chubukov, E. Fradkin, S. A. Kivelson, M. Granath and A. Toschi for useful discussions and suggestions. C.S., L.F., and P.J.H. are supported by the DOE grant number DE-FG02-05ER46236. L.F. acknowledges support by the European Union's Horizon 2020 research and innovation programme through the Marie Skłodowska-Curie grant SuperCoop (Grant No 838526).

## Author contributions
C.S. and L.F. contributed equally to this work. C.S., L.F., and P.J.H discussed and contributed to the project design. C.S. performed the analytical calculation, L.F. performed the numerical analysis. C.S., L.F., and P.J.H. contributed to the data analysis, to the interpretation of the theoretical results, and to the writing of the text.

## Competing interests
The authors declare no competing interests.
