## [Peer Review File · Nature Communications]

Reviewers' comments:

Reviewer #1 (Remarks to the Author):

Report on Microscopic mechanism for fluctuating pair density wave

by Setty, Fanfarilo and Hirschfield.

In this paper, the authors analyze the pair susceptibility of a

superconducting condensate at finite momentum q . The abstract summarizes the work as providing an "analytically solvable model" of pair density waves formed by an anisotropic interaction that "offers a unified microscopic framework that can be used to understand the emergence of fluctuating and long-range pair density waves in realistic systems".

Here are my points about the paper:

(1) The hyperbole in the title and the abstract needs to be toned down. The authors give the impression that the authors have developed a new analytic framework for studying pair density waves in realistic systems. In reality, they have carried out a conventional and approximate analysis of pair fluctuations in a highly simplified BCS model: it is not microscopic. The work has been made within an already well-established, framework; moreover, although I have nothing against BCS model calculations, I think the suggestion that this work provides a unified framework to understand realistic systems as unnecessary hype.

(2) The starting point for the paper is the BCS Hamiltonian with an anisotropic interaction (1) and (2). The authors do not make clear

whether the interaction applies to a narrow shell around the Fermi energy, as in weak-coupling BCS theory, or whether it applies across the entire Fermi surface. Also - not made clear here - are we working in a regime where the Fermi wavevector is much smaller than the inverse lattice constant?

(3) In equation (3), the authors have swept under the rug, the distinction between amplitude and phase fluctuations about a uniform background; even in the normal state Δ_q is a complex valued quantity so that the authors should write $\bar{\Delta}_q \Delta_q$ in (3). However, the text implies that the authors want to consider instabilities of a pre-existing superconducting state, for which (3) is insufficient, even with this correction. A correct treatment of such fluctuations about a

superconducting state is well known, and makes the distinction between the quantity Δ_q and its complex conjugate $\bar{\Delta}_q$. A correct expansion of (3) in the SC state involve a matrix susceptibility with separate eigenvalues for phase and amplitude fluctuations. It involves terms such as $\bar{\Delta}_q \Delta_q$ involving Δ_q and its conjugate, and terms such as $\Delta_{-q} \Delta_q$ and its complex conjugate coupled via a two-by-two matrix. Instabilities into non-uniform states involve negative eigenvalues of this matrix at finite momentum. Unfortunately, the authors gloss over these features in their analysis, writing (3) as if Δ_q is a real quantity.

(4) The authors calculate χ_{c_2} at T_i using equation (4) for extremely large values of $\alpha = E_B/E_F$, and find that it becomes negative,

allowing them to argue for a PDW instability. This is pretty iffy. Although the mean-field equation for the pair susceptibility in equation (4) is formally correct in the normal state at weak coupling, where Δ_0 is set to zero, we must remember that in weak coupling, the presence of logs in the susceptibility, and the fact that $\ln(\Lambda/T_i)$ is large allow us to ignore vertex corrections. Once $\alpha \sim 1$ and the logs are small, we need to account for long lived pair fluctuations which introduce vertex and self-energy corrections. Also, the lattice starts to become very important. In short: is there any reason other than phenomenology, to believe that equation (4) can be used at large values of α ?

I would like to suggest that the authors of this paper work together to rewrite the paper without the hype in the abstract and the title. I

recommend changing the title, removing "Microscopic mechanism" - eg to "Phenomenological model of a fluctuating pair density

wave". The authors can then restrict the treatment of the fluctuations to the normal state, and perhaps, advance phenomenological arguments as to why equation (4) can be used in the BCS-BEC cross-over. The paper should not be

published in its current form.

Reviewer #2 (Remarks to the Author):

The manuscript by Setty et al., analyzes the stability of a homogenous d- or s-wave superconductor under an anisotropic attractive interaction. The topic is important and timely given recent discussions and evidence for pair density waves in the cuprate superconductors, with possible relevance to the elusive pseudogap state.

The authors go beyond standard mean field theory by not only considering the the Ginzburg-Landau instability criterion (the sign of c_0 in eqn 5), but also properly studying the stability towards spatial deformations of the order parameter, i.e. the superfluid stiffness (c_2 in eqn 5). If the stiffness is negative the homogenous sc is unstable. This is certainly an interesting analysis, showing instability in the strong coupling regime and for d-wave pairing symmetry.

However, I'm afraid that the subsequent analysis is not quite correct, and the claim of having established PDW order not substantiated. The authors consider higher order derivatives (eqn 6/fig 4) to indicate stability for a finite momentum mode at $q=\bar{Q}$. This is a relevant indication of a possible finite momentum state, but it is not sufficient to establish stability. The result should be bootstrapped back to solve for the GL stability criterion $c_0(\bar{Q})=0$ in terms of that mode. But that analysis, as far as I can tell, is equivalent to the problem which is consider in ref 11, Loder et al., except for a different dispersion.

The authors would then also need to consider whether to solve for a single mode Δ_Q (Fulde-Ferrell type order) or for two counter propagating modes $\text{PDW}=\Delta_Q+\Delta_{-Q}$. The latter would require an approximation, as it breaks translational invariance, and would also induce charge density wave order. There is certainly more to be said about the stability to spatial deformations of these phases, which was not considered in ref 11. But in order to do that study properly, it should start from the correct mean field state of finite momentum sc.

In addition, even if this issue is be addressed by the authors it is still a mean field study, in the strong coupling regime where it's not clear that MF theory is valid. It is also an effective model with attractive interaction, not a microscopic model, such as the Hubbard or t-J models, for which PDW order in fact has proven difficult to establish. The instability towards spatial deformations of the

homogenous d-wave sc is an interesting observation, but it falls short of the claims made in the paper of providing a “microscopic framework to understand the emergence of ... pair density waves in realistic systems”. If the main concerns are resolved, I think the work would be suitable for a more specialized journal.

Summary of Changes:

1. In regard to comments of referees #1 and #2, we provided an explicit justification of the use of the word microscopic in reference to our model. See discussion below Eq. (2).
2. In regard to comments of referees #1 and #2, we added a discussion about the applicability of weak coupling theory and the non-universal value of the critical α in the conclusions of the manuscript.
3. In regard to comments of referees #1 and #2, we added a section to the Supplementary Material: "Non-universality of the critical α " to provide a clear example of how the value of the critical α is affected by the set of parameters used.
4. In regard to comment of referee #1, we extended the discussion about the Λ parameter and clarified its relation with the lattice parameter as well as with the Fermi energy. See below Eq.(2).
5. In regard to comment of referee #1, we corrected the typo in Eq.(3). Δ_q is indeed the complex Hubbard-Stratonovich field.
6. In regard to comment of referee #2: The statement that we had explicitly tested the stability of the PDW phase by expanding the free energy around $q = \bar{Q}$, previously given in the supplement only, is now summarized also in the main body of the ms (see Eq.(7) and following discussion).

In the following, we address the comments of the referees (black text) point by point (responses in blue text). Some issues have been raised by both referees, relating to the title (and novelty with respect to existing works), as well as validity of our calculations at stronger coupling. Let us therefore start by replying to these important points.

Response to Referees:

General response 1 to all referees: Strength of coupling

Both the referees raised concerns regarding the validity of our study in the strong coupling regime. Here we address this issue.

While in our paper the critical value of α for which c_2 changes sign appears to be of order 1, which is nominally outside the range of standard BCS weak-coupling theory, our result as it stands is a strong indication of a tendency of systems like those we consider to have an instability to a PDW phase. This was also the perspective of the Loder et al and Wårdh et al papers we cited, who argued that this result was interesting enough by itself.

We also note that the critical value of α is a finite non-zero number that does *not* take a universal value. Depending on parameters, e.g. the filling or the cut-off energy Λ , we can easily produce much smaller critical values of α . We provide an example in Fig. 1(a) where we show the behavior of $c_2(\alpha)$ obtained using different values of Λ . As one can see, the only change of the cut-off energy can produce critical value of order 0.3, which might be considered weak-coupling.

FIG. 1. (a) Momentum rigidity parameter c_2 as a function of the coupling strength $\alpha = E_B/E_F$ at $T = T_i$ computed using different values of the cut-off energy Λ . The critical value of α for which c_2 changes signs decreases for larger values of the cut-off energy parameter. (b) Momentum rigidity parameter as a function of the coupling strength $\alpha = E_B/E_F$ at $T = T_i$ for two different q -directions, i.e. \mathbf{q} pointing along x - or y - directions (as shown in the original version of our ms) vs the diagonal axis. The critical value of α is not universal and changes depending the direction of the momentum vector \mathbf{q} .

In addition, we have now shown that for instabilities with \mathbf{q} in the direction of the nodes of the anisotropic interaction, the critical α is much smaller, well within the applicability of weak coupling theories. Hence, there appears to be no physical reason that necessarily constrains FPDW and PDW phases to be in the strong coupling regime of $\alpha \sim 1$.

To show this, we carried out further calculations of the microscopic rigidity parameters by varying the direction of the variable \mathbf{q} in the pair susceptibility. In Fig. 1(b), we make a comparison plot of $c_2(\alpha)$ obtained with \mathbf{q} pointing along x - or y - directions (as shown in the original version of our ms) vs the diagonal axis along the nodes of the d - wave order parameter. We see clearly that the critical α for the sign change of c_2 is much smaller for \mathbf{q} along the diagonals where the critical value is $\alpha \sim 0.25$, well within the applicability of BCS theory. A more thorough phase diagram with \mathbf{q} pointing along arbitrary directions will soon appear in a subsequent work. For the purposes of the current manuscript, we focus on the analysis of \mathbf{q} pointing along the x - or y - directions to enable a comparison with previous numerical literature (e.g. Loder et. al) in which such assumptions have been made.

Note that in the cuprates, the ordering wave vector is along x - or y - direction along the Cu-Cu bonds and not along

the diagonals. The origin of this effect is the subject of much debate and clearly depends on the particular electronic structure of the cuprates, possibly including the van Hove singularity in the density of states. These effects are not considered in our current work and will modify the critical values of α and the direction of \mathbf{q} .

Notice also that fluctuations by competing orders (e.g. charge density wave) have not been taken into account in the present work. These would further affect the momentum rigidity parameter along different \mathbf{q} -directions and, as a consequence, the numerical value of the critical α .

For all the above reasons, we believe that while strong coupling approaches would be useful in confirming our results, they are not essential to demonstrate the proof-of-principle existence of FPDW/PDW phase using our mechanism.

General response 2 to all referees: Use of the word “microscopic”

Both the referees raised concerns regarding whether use of the phrase “microscopic mechanism” is appropriate. Here we address this issue.

We begin by noting that there appears to be substantial differences in opinion in how the word “microscopic” is interpreted in literature and by the community.

In recent years, and particularly within the community discussing the possibility of PDW states in cuprates, the term has been used in connection with the question of whether a “microscopic model”, i.e. a model with repulsive local interactions representing the CuO_2 plane, can be found with a PDW ground state. It appears that the two referees are using the term in this sense, and therefore would say that that BCS or BCS-like models are effective models and a complete microscopic analysis should begin with repulsive Hubbard or $t - J$ models, or similar.

On the other hand, other well established works (as we point out below) consider BCS or BCS-like Hamiltonians as being sufficiently microscopic starting points. In both cases, the theories begin with a full attractive four-fermion interaction.

The difference between the two lies in whether or not one generates from the local repulsion an effective pairing interaction with an attraction in one or more pairing channels, or whether one simply assumes an attraction from the start. In this sense, we agree with the referees that deriving the microscopic origin of the attractive interaction that results in a FPDW/PDW is undoubtedly a more complete solution to the problem, which we have not attempted here. While we were careful not to claim a solution to the microscopic pairing mechanism in the original version of the manuscript, we have made sure to further reiterate this point more explicitly in the new version.

Nevertheless, we feel using the word “microscopic” in the title is justified and consistent with existing literature.

For example, the term “microscopic” was used to differentiate BCS theory from Ginzburg Landau theory in the title of the famous paper *Microscopic derivation of the Ginzburg-Landau equations in the theory of superconductivity* (1959) The original work by Gorkov who provided the first connection between BCS theory and phenomenological model of Ginzburg and Landau. The derivation we provide in our work is an analogous expansion in term of the order parameter “ q ” of the modulated phase starting from a microscopic model.

Another example is provided by the book of Schrieffer, *Theory of Superconductivity* also refers to BCS theory as a microscopic model in the same sense.

Given this context. we wanted to refer to our results as “microscopic” because our model includes minimal *microscopic* ingredients – two particle binding energy E_B , Fermi energy E_F and anisotropic attractive interactions – to explain the common origin of the three important experimentally observed phases of FPDW, PDW and d-wave SC. Such a unified picture is missing in current literature.

Comments of Referee #1:

Referee Comment: In this paper, the authors analyze the pair susceptibility of a superconducting condensate at finite momentum q . The abstract summarizes the work as providing an “analytically solvable model” of pair density waves formed by an anisotropic interaction that “offers a unified microscopic framework that can be used to understand the emergence of fluctuating and long-range pair density waves in realistic systems”.

(1) The hyperbole in the title and the abstract needs to be toned down. The authors give the impression that the authors have developed a new analytic framework for studying pair density waves in realistic systems. In reality, they have carried out a conventional and approximate analysis of pair fluctuations in a highly simplified BCS model: it is not microscopic. The work has been made within an already well-established, framework; moreover, although I have nothing against BCS model calculations, I think the suggestion that this work provides a unified framework to understand realistic systems as unnecessary hype.

Response: We thank the referee for reviewing our paper and providing useful comments. We noticed that the use we have done of the word “microscopic” has been the source of misunderstanding for both reviewers. We provide a general response to both referees (see #2 above) and we revised the manuscript accordingly in order to clarify better what we mean and avoid possible confusion to the reader.

Referee Comment:(2) The starting point for the paper is the BCS Hamiltonian with an anisotropic interaction (1) and (2). The authors do not make clear whether the interaction applies to a narrow shell around the Fermi energy, as in weak-coupling BCS theory, or whether it applies across the entire Fermi surface. Also - not made clear here - are we working in a regime where the Fermi wavevector is much smaller than the inverse lattice constant?

Response: The pairing energy scale of our calculation is Λ , the high energy cut-off set by the inverse lattice spacing. This is the highest energy scale of our problem. We do not restrict our computations to a narrow window of energy since we wish to move away from the extremely weak coupling $\alpha \ll 1$ regime. As we set $\Lambda > E_F$, the Fermi wave vector is also smaller than the inverse lattice constant. We acknowledge the referee’s comments and clarified those key definitions in the revised version of the manuscript.

Referee Comment:(3) In equation (3), the authors have swept under the rug, the distinction between amplitude and phase fluctuations about a uniform background; even in the normal state Δ_q is a complex valued quantity so that the authors should write $\bar{\Delta}_q \Delta_q$ in (3). However, the text implies that the authors want to consider instabilities of a pre-existing superconducting state, for which (3) is insufficient, even with this correction. A correct treatment of such fluctuations about a superconducting state is well known, and makes the distinction between the quantity Δ_q and its complex conjugate $\bar{\Delta}_q$. A correct expansion of (3) in the SC state involve a matrix susceptibility with separate eigenvalues for phase and amplitude fluctuations. It involves terms such as $\bar{\Delta}_q \Delta_q$ involving Δ_q and its conjugate, and terms such as $\Delta_{-q} \Delta_q$ and its complex conjugate coupled via a two-by-two matrix. Instabilities into non-uniform states involve negative eigenvalues of this matrix at finite momentum. Unfortunately, the authors gloss over these features in their analysis, writing (3) as if Δ_q is a real quantity.

Response: We thank the referee for raising this point. We realized the presence of a typo in the Gaussian action displayed in Eq. (3). We indeed performed the Hubbard-Stratonovich transformation accounting for a complex field Δ_q as detailed in the Supplementary material where the reader can also find a complete and correct version of the Gaussian action see Eq. (13). We then fixed the mean-field values of Δ_q to a real value and study fluctuations around that mean-field value of the amplitude.

The referee is right that one could perform the analysis explicitly separating the amplitude and phase sectors. In this case the Gaussian action contains term up to the second order of the phase and amplitude fields instead of $|\Delta|^2$. One could then in principle examine modulated orders in both the phase and amplitude sectors. In this scenario, a negative stiffness in the phase could potentially indicate a modulation of the phase of the order parameter in real space (phase density waves). While such phases are interesting in their own right (which we are currently investigating), the goal of our current work is to demonstrate a minimal model where the pair amplitude is modulated with a finite wave vector since current experiments support such a phase.

Referee Comment: (4) The authors calculate c_2 at T_i using equation (4) for extremely large values of $\alpha = E_B/E_F$, and find that it becomes negative, allowing them to argue for a PDW instability. This is pretty iffy. Although the mean-field equation for the pair susceptibility in equation (4) is formally correct in the normal state at weak coupling, where Δ_0 is set to zero, we must remember that in weak coupling, the presence of logs in the susceptibility, and the fact that $\ln(\Lambda/T_i)$ is large allow us to ignore vertex corrections. Once $\alpha \sim 1$ and the logs are small, we need to account for long lived pair fluctuations which introduce vertex and self-energy corrections. Also, the lattice starts to become very important. In short: is there any reason other than phenomenology, to believe that equation (4) can be used at large values of α ?

Response: As both referees raised similar criticisms about the validity of a weak-coupling calculation we provided a general response above (see #1 above).

Referee Comment: I would like to suggest that the authors of this paper work together to rewrite the paper without the hype in the abstract and the title. I recommend changing the title, removing "Microscopic mechanism" - eg to "Phenomenological model of a fluctuating pair density wave". The authors can then restrict the treatment of the fluctuations to the normal state, and perhaps, advance phenomenological arguments as to why equation (4) can be used in the BCS-BEC cross-over. The paper should not be published in its current form.

Response: We thank the referee for the detailed review of our work that motivated us to revise the manuscript, substantially improving its quality. We address all the referee's comments and overcome the two main criticisms concerning the validity of a weak coupling treatment and the use of the word "microscopic". We provided in this reply and in the new version of the manuscript, additional calculations that further support our treatment and clarified the use of microscopic to describe our model.

Comments of Referee #2:

Referee Comment: The manuscript by Setty et al., analyzes the stability of a homogenous d- or s-wave superconductor under an anisotropic attractive interaction. The topic is important and timely given recent discussions and evidence for pair density waves in the cuprate superconductors, with possible relevance to the elusive pseudogap state.

The authors go beyond standard mean field theory by not only considering the the Ginzburg-Landau instability criterion (the sign of c_0 in eqn 5), but also properly studying the stability towards spatial deformations of the order parameter, i.e. the superfluid stiffness (c_2 in eqn 5). If the stiffness is negative the homogenous sc is unstable. This is certainly an interesting analysis, showing instability in the strong coupling regime and for d-wave pairing symmetry.

Response: We thank the referee for taking time to review our manuscript and providing useful comments to improve our manuscript. We appreciate the referee for recognizing the timeliness of our work and we are glad to see that the referee finds our strategy of studying the stiffness to look for spatial deformation of the order parameter an interesting analysis.

Referee Comment: However, Im afraid that the subsequent analysis is not quite correct, and the claim of having established PDW order not substantiated. The authors consider higher order derivatives (eqn 6/fig 4) to indicate stability for a finite momentum mode at $q = \bar{Q}$. This is a relevant indication of a possible finite momentum state, but it is not sufficient to establish stability. The result should be bootstrapped back to solve for the GL stability criterion $c_0(\bar{Q}) = 0$ in terms of that mode. But that analysis, as far as I can tell, is equivalent to the problem which is consider in ref 11, Loder et al., except for a different dispersion.

Response: We agree with the referee that looking at the higher order derivatives is not enough and that one should look at the GL stability criterion. This is indeed what we have done. The referee may possibly have missed this analysis since it was given in the supplementary information. As one can see from Eqs. (20)-(24) we solved for the GL stability criterion $c_0(\bar{Q}) = 0$ using the new $q = \bar{Q}$ mode perturbatively. With the new solution we found a positive stiffness confirming the stability of the FPDW and PDW states. As noted by the referee this is an important point so we added a summary of this result in the revised version of the manuscript.

We would like to further clarify that our analysis is distinct and goes beyond the Loder et al result in several important ways:

- 1) The calculation by Loder et al is performed at $T = 0$. This is quite a disadvantage since they cannot access the FPDW phase (or the “pseudo-gap equivalent” of the PDW phase) which is the main theme of our work. This phase is new since, to our knowledge, we are not aware of a analytically solvable microscopic model that leads to formation of finite momentum Cooper pairs *without* long-range order. Such a phase must invariably exist in the relevant moderate to strong coupling limit and at finite temperatures.
- 2) In our model, the coefficients of both the q^2 and q^4 terms are negative but that of the q^6 term is positive – this means there are additional phase distinctions one can made depending on the GL coefficients. See for example the phase diagram in Fig 2 of Ref. 20 that cannot be accessed via Loder et al analysis.
- 3) Our model provides a clear physical picture of the origin of the FPDW/PDW phases in terms of d-wave fluctuations. This is made possible because our model is analytically solvable unlike the numerical results of Loder. Our work determines an expression for the critical pairing strength, α , in terms of μ , that shows how c_2 changes sign as μ becomes negative. This is in agreement with our numerics as one can see from inset of Fig. 3 (a).
- 4) Our strategy i.e., to look at the sign of c_2 to identify a possible FPDW/PDW state, allows us to unveil the role of the pairing anisotropy in establishing the modulated phase. In fact, as we understand from Fig. 5, the s-wave isotropic pair fluctuations cannot yield modulated phases, while for the d-wave case, the negative contribution to the c_2 parameter comes from the nodal regions of the Brillouin zone. This insight is novel and cannot be addressed by other numerical studies like Loder et al.

5) Furthermore, our calculations clarify the role played by the lattice showing how the modulated phases vanish in the continuum limit. This picture would not have been possible if not for the analytical tractability of model where the physics is made more transparent.

Referee Comment: The authors would then also need to consider whether to solve for a single mode Δ_Q (Fulde-Ferrell type order) or for two counter propagating modes $PDW = \Delta_Q + \Delta_{-Q}$. The latter would require an approximation, as it breaks translational invariance, and would also induce charge density wave order. There is certainly more to be said about the stability to spatial deformations of these phases, which was not considered in ref 11. But in order to do that study properly, it should start from the correct mean field state of finite momentum sc.

Response: Our current work focuses on the relationship between FPDW/PDW/dSC orders and neglects the role of charge order. This is because we only take into account the electron interactions in the Cooper channel. We leave out a more serious treatment of the charge order with counter-propagating pairing modes for future work.

Referee Comment: In addition, even if this issue is be addressed by the authors it is still a mean field study, in the strong coupling regime where its not clear that MF theory is valid. It is also an effective model with attractive interaction, not a microscopic model, such as the Hubbard or t-J models, for which PDW order in fact has proven difficult to establish. The instability towards spatial deformations of the homogenous d-wave sc is an interesting observation, but it falls short of the claims made in the paper of providing a microscopic framework to understand the emergence of pair density waves in realistic systems. If the main concerns are resolved, I think the work would be suitable for a more specialized journal.

Response: Similar concerns about the validity of our treatment at strong coupling and the use of the word "microscopic" were also raised by the other referee so we decided to provide a general response. The referee can find our reply to the comments:

- "even if this issue is be addressed by the authors it is still a mean field study, in the strong coupling regime where its not clear that MF theory is valid" in the general response #1
- "It is also an effective model with attractive interaction, not a microscopic model, such as the Hubbard or t-J models, for which PDW order in fact has proven difficult to establish" in response #2.

We thank the referee for the detailed review of our work that motivated us to revise the manuscript, substantially improving its quality. We address all the referee's comments and overcome the two main criticisms concerning the validity of a weak coupling treatment and the use of the word "microscopic". We provided in this reply and in the new version of the manuscript, additional calculations that further support our treatment and clarified the use of microscopic to describe our model.

REVIEWER COMMENTS

Reviewer #1 (Remarks to the Author):

Second Report on Microscopic mechanism for fluctuating pair density wave
by Setty, Fanfarillo and Hirschfeld.

(1) In my original report, I wrote that

"The hyperbole in the title and the abstract needs to be toned down. The authors give the impression that the authors have developed a new analytic framework for studying pair density waves in realistic systems. In reality, they have carried out a conventional and approximate analysis of pair fluctuations in a highly simplified BCS model: it is not microscopic. The work has been made within an already well-established, framework; moreover, although I have nothing against BCS model calculations, I think the suggestion that this work provides a unified framework to understand realistic systems as unnecessary hype."

In their response, the authors argue that a precedent for their use of "Microscopic" was set by BCS and Gor'kov in the 1950s. Yet in BCS theory they did not simply introduce a pair interaction without first relating it to the microscopic physics of phonon exchange and considering (with Pines) the microscopic effects of the Coulomb interaction, neglected in this paper despite their importance for pair fluctuations. Moreover, since the Eliashberg theory of the 1960s, we would no longer describe BCS theory as microscopic - a point that is emphasized in the famous two-volume by Parks. I think the

editors should take note that both referees felt very strongly that this is not a microscopic theory. I stand by my original comment. Publication of a this theory advertised as a "microscopic mechanism" should not occur.

(2) On the second point, I wrote

"The starting point for the paper is the BCS Hamiltonian with an anisotropic interaction (1) and (2). The authors do not make clear whether the interaction applies to a narrow shell around the Fermi energy, as in weak-coupling BCS theory, or whether it applies across the entire Fermi surface. Also - not made clear here - are we working in a regime where the Fermi wavevector is much smaller than the inverse lattice constant?"

The authors have clarified this point.

(3) The authors have still not correctly calculated the fluctuation susceptibility for fluctuations in the superconducting state. In my original report, I wrote

"In equation (3), the authors have swept under the rug, the distinction between amplitude and phase fluctuations about a uniform background; even in the normal state Δ_q is a complex valued quantity so that the authors should write $\bar{\Delta}_q \Delta_q$ in (3). However, the text implies that the authors want to consider instabilities of a pre-existing superconducting state, for which (3) is insufficient, even with this correction. A correct treatment of such fluctuations about a superconducting state is well known, and makes the distinction between the quantity Δ_q and its complex

conjugate $\bar{\Delta}_q$. A correct expansion of (3) in the SC state involve a matrix susceptibility with separate eigenvalues for phase and amplitude fluctuations. It involves terms such as $\bar{\Delta}_q \Delta_q$ involving Δ_q and its conjugate, and terms such as $\Delta_{-q} \Delta_q$ and its complex conjugate coupled via a two-by-two matrix. Instabilities into non-uniform states involve negative eigenvalues of this matrix at finite momentum. Unfortunately, the authors gloss over these features in their analysis, writing (3) as if Δ_q is a real quantity."

Although the authors have corrected the typo in equation (3), the expression that they give for the fluctuations in equation (11) is still incorrect, for it still does not properly consider the coupled fluctuations between Δ_q and Δ_{-q} . The authors don't seem to understand that Δ_q and Δ_{-q} are linearly coupled, separate Gaussian variables. The correct expression, when you include the couplings, involves two fluctuation modes: both must be considered in the Gaussian fluctuations, as both feature inside the path integral. This leads to two eigenvalues for Π_q , a symmetric one, one involving amplitude fluctuations, and an antisymmetric one, involving phase fluctuations, with $\Pi^{\pm}(q) = \sum (GG \pm FF)$. The authors have effectively assumed that their fluctuations are real, and they have correctly given the result for the symmetric eigenmode, but they have not given the equally important mode for the phase fluctuations. It is mathematically incorrect to select one mode over the other. Even worse to entirely omit the second one. The authors will have to decide whether they are talking about a phenomenologically neutral model, in which the transverse mode is a Boguilubov mode at $q=0$, or whether they are dealing with a microscopic model, which acknowledges the

electromagnetic/Coulomb interaction that introduces plasmons into all the long-wavelength modes.

These are issues that were correctly treated in the classic works of the 1950s: there is no excuse for omitting them 70 years later.

The paper can not be published without this oversight being corrected with an appropriate discussion, both in the paper, and in the supplementary materials.

(4) I previously wrote: "The authors calculate ϵ_{c_2} at T_i using equation (4) for extremely large values of $\alpha = E_B/E_F$, and find that it becomes negative,

allowing them to argue for a PDW instability. This is pretty iffy. Although

the mean-field equation for the pair susceptibility in equation (4) is

formally correct in the normal state at weak coupling, where Δ_0 is set

to zero, we must remember that in weak coupling, the presence of logs in the

susceptibility, and the fact that $\ln(\lambda/T_i)$ is large allow us to

ignore vertex corrections. Once $\alpha \sim 1$ and the logs are small, we

need to account for long lived pair fluctuations which introduce vertex and

self-energy corrections. Also, the lattice starts to become very important. In

short: is there any reason other than phenomenology, to believe that equation

(4) can be used at large values of α ?"

In their reply, the authors claim that equation (4) can be used as a

"tendency" towards a PDW instability. I don't know how they know this, especially in a theory that completely ignores the Coulomb interaction. The authors should come clean and state this as an assumption of the theory.

In conclusion, the authors haven't adequately addressed

the scientific concerns that have been highlighted by their referees.

The problem is, they are caught between a rock and a

hard place - on the one hand, they insist that their theory is microscopic, yet their treatment is phenomenological and omits some important physics. I would like to once again suggest that the authors of this paper work together to correct the paper without the hype in the abstract and the title, removing "Microscopic mechanism" from the title. The authors also need to do a better job treating the fluctuations of the complex gap function, and they need to state clearly that they have chosen to completely omit the Coulomb interaction - or better - to include it. As it stands, the paper is not ready for publication.

Reviewer #2 (Remarks to the Author):

I think the authors have clarified several of the caveats of my first review. It is a quite striking phenomenon that is observed, that with a suitable form factor of the interaction a finite momentum pairing state might arise at not very strong coupling. As discussed already in the first review this is a very important problem in the community of strongly correlated materials. However, I think the presentation needs to be made clearer in several places. If the authors can address these questions I would recommend the paper for publication.

1) What type of state is actually explored. It's discussed as a PDW state, but it's only a single component which is solved for. (As discussed already in my first review.) The state would thus be a Fulde-Ferrell state that breaks time reversal. A PDW would normally have both components, and breaks translational invariance. In order to make the distinction, quartic terms in the pair fields need to be considered. Even though this may be beyond the scope of the present paper, it should be clarified in the text.

2) If I understand correctly the authors do solve for the mean field pair field, not only T_c . For this the quartic field terms are needed. I don't see those anywhere in the presentation. To make the presentation more clear it would be interesting to also show the Fermi surfaces. Also to calculate the current as a function of q , at the FF local minimum $q=Q$ and away from that. The Free energy has a minimum at Q so this should correspond to a zero current state, but to actually calculate the current based on the gap and the BCS (FF) wave function (as done in reference 19) would be an independent confirmation that the physics is consistent.

3) Fluctuating phase of the PDW. This is just an extrapolation? Please clarify what in the phase diagram is based on calculations, and what is speculation.

4) Angular dependence. Bond aligned or diagonal order. If there are results showing that the diagonal order is more stable please show this. In a PDW state these would correspond to different representations of broken C_4 symmetry, B_{1g} or B_{2g} . It may be interesting to refer to recent work, where both types of nematic order (which could be the "fluctuating PDW phase") are found: Wårdh, J., Granath, M., Wu, J., Bollinger, A. T., He, X., & Božović, I. (2022). Colossal transverse magnetoresistance due to nematic superconducting phase fluctuations in a copper oxide. arXiv preprint arXiv:2203.06769.

5) There seems to be a strong dependence of the value of Q on the parameters of the model, especially the cut-off. Given a parabolic band that the authors are using, the cut-off introduces an effective band-edge. How does the Q depend on that, and are the results robust to instead doing the calculations on a lattice with a Brillouin zone.

Dear Editor,

We would like to resubmit the manuscript NCOMMS-21-43909-T with the new title “Mechanism for fluctuating pair density wave” to Nature Communications. We thank the two referees for assessing the manuscript, and for providing guidance on how to improve the paper.

The following material is organized as:

- a. Our point-by-point response to the individual referee reports
- b. Summary of Changes

We hope that the current version of the manuscript addresses all the recommendations and critiques of the referees.

Best regards,

The authors

Chandan Setty, Laura Fanfarillo and Peter Hirschfeld

Response to Referee 1:

We would like to begin by apologizing to you for the long delay between receipt of the latest round of reports and the resubmission. To some extent this was occasioned by the rethinking required by the good points you raised, and to some extent by personal circumstances of some of the authors making it difficult to focus on this project.

Referee Comment: In their response, the authors argue that a precedent for their use of "Microscopic" was set by BCS and Gor'kov in the 1950s. Yet in BCS theory they did not simply introduce a pair interaction without first relating it to the microscopic physics of phonon exchange and considering (with Pines) the microscopic effects of the Coulomb interaction, neglected in this paper despite their importance for pair fluctuations. Moreover, since the Eliashberg theory of the 1960s, we would no longer describe BCS theory as microscopic - a point that is emphasized in the famous two-volume by Parks. I think the editors should take note that both referees felt very strongly that this is not a microscopic theory. I stand by my original comment. Publication of a this theory advertised as a "microscopic mechanism" should not occur.

Response: We understand and appreciate the referee's concern that our work does not begin from repulsive Coulomb interactions to derive a modulated pairing solution. In effect, we have neglected the role of Coulomb interactions on the phase diagram. In the new version of the manuscript, we have followed the referee's advise and removed the phrase "Microscopic mechanism" from the title of the paper. Instead, we have replaced it with "Mechanism for fluctuating pair density wave".

Referee Comment: Although the authors have corrected the typo in equation (3), the expression that they give for the fluctuations in equation (11) is still incorrect, for it still does not properly consider the coupled fluctuations between Δ_q and Δ_{-q} . The authors don't seem to understand that Δ_q and Δ_{-q} are linearly coupled, separate Gaussian variables. The correct expression, when you include the couplings, involves two fluctuation modes: both must be considered in the Gaussian fluctuations, as both feature inside the path integral. This leads to two eigenvalues for Π_q , a symmetric one, one involving amplitude fluctuations, and an antisymmetric one, involving phase fluctuations, with $Pi^\pm(q) = \sum(GG \pm FF)$. The authors have effectively assumed that their fluctuations are real, and they have correctly given the result for the symmetric eigenmode, but they have not given the equally important mode for the phase fluctuations. It is mathematically incorrect to select one mode over the other. Even worst to entirely omit the second one.

Response: We thank the Referee for explaining more their concern. We realized that we omitted some important details of the derivation in the previous version of the manuscript. We provide extensive response to the Referee criticism below and revised the manuscript accordingly.

In our Hubbard-Stratonovich treatment we consider a complex HS field Δ . This is clear if one looks at the set of equations displayed in the Supplementary material, Eq.s (1-7). As the Referee correctly pointed out our theory, dealing with a complex order parameter, contains two degrees of freedom and that their Gaussian fluctuations are associated with the symmetric and antisymmetric combination of the Green functions i.e. $Pi^\pm(q) \sim \sum(GG \pm FF)$.

The Referee concern comes from the fact that we did not consider both modes and instead focus explicitly on the fluctuations of the Real part of the HS field. In Eq.13, in fact, we specify the expression for $Pi(q)$ using only the combination associated to the real part of the fluctuating field. Notice, however, that when approaching the transition from above $T \rightarrow T_i^+$, it is meaningless to talk about modulus and phases (or real and imaginary part) of the fluctuations. In fact, at T_i , the Gaussian propagator for both modes reduces simply to the product of the normal Green function as $FF = 0$. The main result of our work is the appearance of a tendency toward a PDW state as signaled by the sign change of c_2 close to the instability temperature. This is obtained by momentum expansion of the $Pi(q) \sim \sum GG$ propagator, thus it is clear that no approximation has been done here and that the result is robust.

Moving to the analysis of fluctuation below T_i , we agree that both real and imaginary part of the complex field should be accounted for. A full calculation of the fluctuations of the complex pairing field in the broken phase can be found for example in Appendix A of Marciani et al Phys. Rev. B 88 2013. In Eq. A5 of this paper the Gaussian action for (ReDelta, ImDelta) is explicitly shown. From Eq.s A5-A7 we can recognize the symmetric and antisymmetric expressions of the propagators controlling the quadratic fluctuations of $Re/Im\Delta$. We also find an off-diagonal term that coupled the two degrees of freedom. By explicit calculation of the mixed bubble it can be shown that the off-diagonal element of the gaussian action vanishes in the static limit, leading to an effective decoupling between real and imaginary part of fluctuations. Since our calculation is performed in the static limit, we can assume

fluctuations of the real and imaginary part of Δ decoupled. We focus then on the analysis of a possible modulated state of the amplitude of the order parameter only. Notice that, due to the effective decoupling of the real and imaginary part of the fluctuating mode in the static limit, the expression for the susceptibility associated to $|Re\Delta|^2$ would not change in the presence of the imaginary counterpart. As a consequence, even in the broken phase, the analysis of the momentum rigidity parameter c_2 extracted from the gaussian susceptibility associated to $|Re\Delta|^2$ is robust, regardless the inclusion of the imaginary part of the fluctuations.

In the revised version of the manuscript we better explain our procedure. We discuss the presence of two independent eigenvalues of $\Pi(q)$ associated to the complex order parameter and justify our choice of focusing on the fluctuations of $Re\Delta$ only. In the supplementary material we write explicitly Eq. 13 for the real and imaginary part of the fluctuations. We show explicitly that while at T_i the two reduce to the same form, below T_i the susceptibilities are different and the two modes are effectively decoupled only in the static limit.

Referee Comment: The authors will have to decide whether they are talking about a phenomenologically neutral model, in which the transverse mode is a Boguilubov mode at $q=0$, or whether they are dealing with a microscopic model, which acknowledges the electromagnetic/Coulomb interaction that introduces plasmons into all the long-wavelength modes. These are issues that were correctly treated in the classic works of the 1950s: there is no excuse for omitting them 70 years later. In their reply, the authors claim that equation (4) can be used as a "tendency" towards a PDW instability. I don't know how they know this, especially in a theory that completely ignores the Coulomb interaction. The authors should come clean and state this as an assumption of the theory.

Response:

We agree with the Referee that the inclusion of the long-range Coulomb interaction is necessary crucial to analyze collective modes of the order parameter. Such inclusion has profound effects on the fluctuations of the pairing field and this reflects in the calculation of physical quantities, e.g. the current response. We believe, however that in our analysis the inclusion of long-range Coulomb interaction would not change our conclusions in the case of a charged system.

As we discussed above, our results are derived from the analysis of the pairing susceptibility in the static limit, in which fluctuations of the real and imaginary part of the pairing field are decoupled. As we do not take into account fluctuations of the phase mode, which is the one that strongly couples to long-range Coulomb interaction, we are confident that the inclusion of these interactions would not affect qualitatively our results. In a charged system the phase mode affects dynamics at the scale of the plasma frequency. The only situation of which we are aware when the phase mode renormalizes *static* quantities is when the system is anisotropic. For example in Ref. (Phys. Rev. Lett. 56, 2513 (1986)), there is an explicit static "backflow" contribution to the stiffness. The fate of this term in the charged system is discussed in Phys. Rev. B 47, 8837 (1993). The backflow term represents the off-resonance excitation of the collective mode implicit in the early studies of gauge invariance in the pairing theory by Rickayzen, Anderson and others, which are presumably the works the referee has in mind. Note, however, that in the static *isotropic* case the corrections to the stiffness from these terms vanish. Since we study instability of a homogeneous *d*-wave ground state, this is equivalent to an isotropic case since the 2nd rank response tensor is isotropic. Thus we conclude that inclusion of the long-range Coulomb interaction would a priori not affect our results here. We have made a comment on these issues in the supplementary information.

Referee Comment: In conclusion, the authors haven't adequately addressed the scientific concerns that have been highlighted by their referees. The problem is, they are caught between a rock and a hard place - on the one hand, they insist that their theory is microscopic, yet their treatment is phenomenological and omits some important physics. I would like to once again suggest that the authors of this paper work together to correct the paper without the hype in the abstract and the title, removing "Microscopic mechanism" from the title. The authors also need to do a better job treating the fluctuations of the complex gap function, and they need to state clearly that they have chosen to completely omit the Coulomb interaction - or better - to include it. As it stands, the paper is not ready for publication.

Response: We thank the Referee for raising these important issues that made us clear that some aspects of our methodology and analysis were not enough discussed or justified. We properly addressed each criticism and revised the paper accordingly in order to make the manuscript suitable for publication.

Response to Referee 2:

We would like to begin by apologizing to you for the long delay between receipt of the latest round of reports and the resubmission. To some extent this was occasioned by the rethinking required by the good points you raised, and to some extent by personal circumstances of some of the authors making it difficult to focus on this project.

Referee Comment: I think the authors have clarified several of the caveats of my first review. It is a quite striking phenomenon that is observed, that with a suitable form factor of the interaction a finite momentum pairing state might arise at not very strong coupling. As discussed already in the first review this is a very important problem in the community of strongly correlated materials. However, I think the presentation needs to be made clearer in several places. If the authors can address these questions I would recommend the paper for publication.

Response: We thank the referee for recognizing the importance of our results. In the new version of the manuscript, we have improved the presentation and addressed the questions of the referee. We are hopeful that this revised version of the manuscript is deemed suitable for publication in Nature Communications.

Referee Comment: 1) What type of state is actually explored. It's discussed as a PDW state, but it's only a single component which is solved for. (As discussed already in my first review.) The state would thus be a Fulde-Ferrell state that breaks time reversal. A PDW would normally have both components, and breaks translational invariance. In order to make the distinction, quartic terms in the pair fields need to be considered. Even though this may be beyond the scope of the present paper, it should be clarified in the text.

Response: We agree with the Referee that in order to be able to distinguish between a PDW and a FF state we would need to analyze the action up to the fourth order in the pairing field. In the introduction of the revised version of the manuscript we emphasize the Gaussian approximation characterizing our analysis and explicitly mention the limitation of our study.

Referee Comment: 2) If I understand correctly the authors do solve for the mean field pair field, not only T_c . For this the quartic field terms are needed. I don't see those anywhere in the presentation. To make the presentation more clear it would be interesting to also show the Fermi surfaces. Also to calculate the current as a function of q , at the FF local minimum $q=Q$ and away from that. The Free energy has a minimum at Q so this should correspond to a zero current state, but to actually calculate the current based on the gap and the BCS (FF) wave function (as done in reference 19) would be an independent confirmation that the physics is consistent.

Response: We thank the Referee for raising this issue and allow us to clarify a potential source of misunderstanding.

In our analysis we do solve the equation for T_c as well as the mean field pairing field, however this is done for the homogeneous solution only. We use perturbative approach in which studying the momentum dependence of the gaussian fluctuations around the homogeneous solution we highlight a tendency toward a PDW state. We do not solve the mean field equations for the finite momentum pairing field $\Delta(q)$. This is why the Referee would not find the quartic fields term in the manuscript. We revised the text to better explain our procedure.

Regarding the calculation of the current as a function of q , we agree that that would be a nice independent confirmation that the physics discussed using the free energy is consistent. For the current calculation, it is convenient to use the finite momentum wave function ansatz for the chosen interactions. This was only done in our latest preprint arXiv:2209.10568. We think this calculation could be addressed in such a non-perturbative setting and suitable to add in a new version of this new paper.

Referee Comment: 3) Fluctuating phase of the PDW. This is just an extrapolation? Please clarify what in the phase diagram is based on calculations, and what is speculation.

Response: $T = T_i$ is the instability temperature of the homogeneous pairing field obtained within the mean-field approximation. $T = T^*$ is the temperature at which the finite momentum fluctuations around the homogeneous solution become critical and it is computed as the temperature at which the rigidity momentum parameter c_2 changes

sign. The $T = T_c$ line is instead the result of speculations based on the analysis of pairing fluctuations in s-wave superconductors. In principle it could be computed within a fluctuations analysis that treats separately the modulus and phase of the order parameter. In the new version of the manuscript, we have better clarified the parts of the phase diagram that have been derived versus the parts that are speculated.

Referee Comment: 4) Angular dependence. Bond aligned or diagonal order. If there are results showing that the diagonal order is more stable please show this. In a PDW state these would correspond to different representations of broken C_4 symmetry, B_{1g} or B_{2g} . It may be interesting to refer to recent work, where both types of nematic order (which could be the “fluctuating PDW phase”) are found: Wardh, J., Granath, M., Wu, J., Bollinger, A. T., He, X., & Bozovic, I. (2022). Colossal transverse magnetoresistance due to nematic superconducting phase fluctuations in a copper oxide. arXiv preprint arXiv:2203.06769.

Response: We thank the referee for pointing to these references. We have included and discussed them in the new version of the manuscript. Concerning the angular dependence of the finite momentum fluctuations, in light of our recent manuscript arXiv:2209.10568, we checked our preliminary results presented in our previous rebuttal regarding the stability of bond aligned vs diagonal order. After performing a more accurate analysis, we realized that the preliminary estimate we offered in our previous response contained a mistake. In the revised calculation, although it is still true that the two modes behave differently, the finite momentum fluctuations along x/y remain the more stable at any temperature for a B_{1g} form factor consistent with arXiv:2209.10568. To obtain a diagonal pair density wave order in our analysis, the interaction must contain a dominant B_{2g} pair fluctuation interaction. This would imply that the form factor of the kind $\sin k_x \sin k_y$ would be dominant in the fluctuations instead of the $\cos k_x - \cos k_y$ form factor. With this form factor, our result continues to hold but with the dominant instability now occurring along the diagonals. Note, however, that our theory does not justify which of the two (B_{1g} or B_{2g}) symmetry channels are the dominant fluctuations. In either of the two cases, one can obtain the finite momentum pairing instability without requiring a strong coupling treatment of the theory.

Referee Comment: 5) There seems to be a strong dependence of the value of Q on the the parameters of the model, especially the cut-off. Given a parabolic band that the authors are using, the cut-off introduces an effective band-edge. How does the Q depend on that, and are the results robust to instead doing the calculations on a lattice with a Brillouin zone.

Response: We thank the referee for raising these interesting questions. In fact, we recently addressed the dependence of the finite Q solution on the cut-off as well as the presence of a lattice Brillouin zone in a recent preprint arXiv:2209.10568 with additional co-authors. Here we showed that a) in the continuum, the homogeneous solution becomes unstable for any finite cut-off provided the interaction is above a critical value. This statement is independent of the cut-off, and b) however, on a lattice, the cut-off indeed plays an important role. Without the cut-off in the interaction, the modulated solution on a lattice loses to the homogeneous solution. We have alluded to these results in the new version of the manuscript.

Summary of changes:

- We changed the title in “Mechanism for fluctuating pair density wave”
- In the supplementary material, we explicitly discussed the presence of two fluctuating modes associated to the complex order parameter and justify our choice of focusing on the fluctuations of $\text{Re}\Delta$ only.
- In the introduction we add a sentence about the limitation of our Gaussian approach “Since we study the action only to second order in the pair field we cannot distinguish between an instability to a FF or LO state.”
- We revised the text relative to the description of the phase diagram of Fig.1 to clarify that T_i is a computed transition temperature for the homogeneous d-wave superconductors, while T_c is not computed but simply deduced from analogous study for the s-wave superconductors in the strong coupling limit.
- We discuss the role of the cut-off in the Conclusion section of the manuscript. See paragraph “Furthermore, we note the key role of the existence of a lattice momentum cut-off ...”
- We added a paragraph in the Conclusion section to discuss the possible connection with recent experiments on B_{2g} PDW in cuprates. See paragraph “Recently Refs. [58, 59] discussed the possible existence of a diagonal pair density wave order in the cuprates ...”

REVIEWERS' COMMENTS

Reviewer #1 (Remarks to the Author):

The authors have made considerable progress in their response to the referees. In their rebuttal the authors do acknowledge that phase fluctuations have been dropped. This needs to be made clear in the main text. In particular:

(1) In their response to Item II. of my previous report, the authors have acknowledged that they have dropped the phase fluctuations in their analysis. However on page 3 the narrative states that

"we instead analyze the gaussian action obtained by retaining up to the second order in the fluctuating field".

This is false. This statement should also add (in their own words)

"In our analysis, we omit the Gaussian fluctuations in the phase of the order parameter."

As it stands, the main text of the paper creates a false impression that all Gaussian fluctuations have been included. Half of them have been dropped.

(2) Equation (3) has not been modified to reflect the discussion in the SM. Here the authors should clearly explain that the expression provided in equation (3) refers only to the Gaussian amplitude fluctuations. There is another contribution to the action that has been omitted. The author has cunningly put the subscript in momentum outside the modulus sign, which may be a subtle way of stating this, but this will be missed by the casual reader! The authors need to make this clear. I suggest adding a statement of the form

"Here we have omitted the contribution of the Gaussian phase fluctuations of the order parameter to the action"

after equation (5).

(3) The authors claim in the abstract that the work provides a

"unified microscopic framework to understand the emergence of both fluctuating and long range pair density waves in realistic systems"

In the light of the above discussion, I recommend dropping this statement from the abstract.

Summary: The current work by Setty, Fanfarillo and Hirschfeld is an incomplete phenomenology, because it omits, without discussion, the phase fluctuations of the order parameter. These degrees of freedom describe the current fluctuations that accompany a pair density wave and as such, they are an important part of the physics.

While they certainly decouple from the amplitude modes at long wavelengths, density waves involve a modulated structure which necessarily involves short-wavelength physics, and for this reason it seems reasonable that phase fluctuations should be included in a complete theory. These fluctuations play a central role in the Meissner screening and they may make an important contribution to the total free energy, modifying the phase diagram. They may also be important in the development of circulating current phases. I hope that future improvements of the current work will take these into account.

Conditional on the three changes itemized above, I recommend this paper for publication.

Reviewer #2 (Remarks to the Author):

I believe the authors have adequately addressed the issues raised in the previous round of review. The observation that a d-wave form factor can give an instability of the homogenous superconductor to a finite wavelength modulated state seems sound, and potentially important. I recommend that the paper should be accepted.

The following material is organized as:

- a. Our point-by-point response to the individual referee reports.
- b. Summary of Changes (appearing in red text in the main manuscript).

Response to Referee 1:

Referee Comment: (1) In their response to Item II. of my previous report, the authors have acknowledged that they have dropped the phase fluctuations in their analysis. However on page 3 the narrative states that

“we instead analyze the gaussian action obtained by retaining up to the second order in the fluctuating field.” This is false. This statement should also add (in their own words) “In our analysis, we omit the Gaussian fluctuations in the phase of the order parameter.” As it stands, the main text of the paper creates a false impression that all Gaussian fluctuations have been included. Half of them have been dropped.

Response: We thank the referee for pointing out this suggestion and we have modified the text in the revised version accordingly (see new paragraph after Eq. 3).

Referee Comment:(2) Equation (3) has not been modified to reflect the discussion in the SM. Here the authors should clearly explain that the expression provided in equation (3) refers only to the Gaussian amplitude fluctuations. There is another contribution to the action that has been omitted. The author has cunningly put the subscript in momentum outside the modulus sign, which may be a subtle way of stating this, but this will be missed by the casual reader! The authors need to make this clear. I suggest adding a statement of the form ”Here we have omitted the contribution of the Gaussian phase fluctuations of the order parameter to the action” after equation (5).

Response: We revised the manuscript to make clear the approximation used here (see new paragraph after Eq. 3)

Referee Comment: (3) The authors claim in the abstract that the work provides a ”unified microscopic framework to understand the emergence of both fluctuating and long range pair density waves in realistic systems” In the light of the above discussion, I recommend dropping this statement from the abstract.

Response: We have now removed the word “microscopic” from the last sentence in the abstract as was previously recommended by the referee. That our results provide a unified framework to understand the emergence of both fluctuating and long range pair density waves is a key aspect of our work.

Referee Comment: Summary: The current work by Setty, Fanfarillo and Hirschfeld is an incomplete phenomenology, because it omits, without discussion, the phase fluctuations of the order parameter. These degrees of freedom describe the current fluctuations that accompany a pair density wave and as such, they are an important part of the physics. While they certainly decouple from the amplitude modes at long wavelengths, density waves involve a modulated structure which necessarily involves short-wavelength physics, and for this reason it seems reasonable that phase fluctuations should be included in a complete theory. These fluctuations play a central role in the Meissner screening and they may make an important contribution to the total free energy, modifying the phase diagram. They may also be important in the development of circulating current phases. I hope that future improvements of the current work will take these into account.

Conditional on the three changes itemized above, I recommend this paper for publication.

Response: We are glad that the referee recommends our manuscript for publication upon the condition that we have made the above changes. We have made the changes suggested by the referee; in particular, we have explicitly discussed the decoupling of phase fluctuations in the main text (after Eq. 3) and SM, and removed the word “microscopic” from the last sentence of the abstract. We are hopeful that the revised version is suitable for final acceptance in Nature Communications. We thank the referee for several useful suggestions that improved our paper.

Response to Referee 2:

Referee Comment: I believe the authors have adequately addressed the issues raised in the previous round of review. The observation that a d-wave form factor can give an instability of the homogenous superconductor to a finite wavelength modulated state seems sound, and potentially important. I recommend that the paper should be accepted.

Response: We are delighted that the referee recommends our paper for publication in Nature Communications. We thank the referee for several useful suggestions that improved our paper.

Summary of changes:

- In regard to point (1) and (2) of Referee 1, we added a paragraph below Eq. (3) to clarify that we have neglected the phase fluctuations of the order parameter.
- In regard to point (3) of Referee 1, we have modified the last line of the abstract to “unified framework to understand the emergence of both fluctuating and long range pair density waves in realistic systems” by removing the word “microscopic”.
- Added a new citation to Ref. 61